# Modeling Collision Probability on Freeway: Accounting for Different Types and Severities in Various LOS

**Bo Yang** [1], **Yao Wu** [1,2,*], **Weihua Zhang** [3] **and Jie Bao** [4]

1   Jiangsu Key Laboratory of Urban ITS, Jiangsu Province Collaborative Innovation Center of Modern Urban Traffic Technologies, Southeast University, Nanjing 210096, China; ybseu2013@126.com
2   Department of Civil Engineering, The University of British Columbia, Vancouver, BC V6T 1Z4, Canada
3   School of Automobile and Traffic Engineering, Hefei University of Technology, Hefei 230009, China; weihuazhang@hfut.edu.cn
4   Civil Aviation College, Nanjing University of Aeronautics and Astronautics, Nanjing 211106, China; jiebao@nuaa.edu.cn
*   Correspondence: wuyaomaster@126.com

**Abstract:** In this study, collision-related data were collected on the I-880 freeway of California in the United States from 2006 to 2011. Our objective was to study the collision probability of different collision types and severities in different traffic states. The traffic states were divided by the traditional level of service (LOS) method. Various Bayesian conditional logit models have been established to analyze the relationship between the collision probability of different collision patterns and LOSs. The results showed that LOS A had the best safety performance associated with all of the collision types and severities, LOS C had the worst safety performance associated with hit object collisions, LOS D had the worst safety performance associated with sideswipe collisions and rear end collisions, and LOS F had the worst safety performance associated with injury collisions. The five-stage Bayesian random parameter sequential logit model was established to quantify the effects of different variables on the collision probability of various collision types and severities. In addition to LOS, the visibility, road surface, weather, ramp, and number of lanes had significant effects on different collision types and severities.

**Keywords:** freeway; safety; LOS; collision types and severities; conditional logit models; Bayesian approach; sequential logit model

## 1. Introduction

With the widespread use of freeway traffic surveillance systems, researchers have started using high-resolution dynamic traffic flow data to identify traffic conditions before collision occurrences. Numerous studies have developed real-time collision probability models for estimating the relative probability of collisions, given dynamic traffic flow data [1–5]. These studies have generally used a case-controlled study design structure, in which the traffic conditions before collisions were considered collision cases, while those under collision-free conditions were considered control cases. With the case-controlled dataset, researchers have developed real-time collision probability models to analyze the relationship between the probability of a collision and the traffic-related variables, including geometric design factors, environment factors, traffic flow factors, crash characteristic factors, driver behavior factors, and control strategy factors on a freeway.

Numerous researchers have studied the spatio-temporal evolution of traffic flows by dividing the traffic flows into different states. However, relatively few studies have investigated the collision types

under different traffic conditions. Thus, it is necessary to explore the collision mechanism of different types and severities in various traffic flow states. In previous studies, it has been proven that there is a significant difference in safety performance for different collision types and severities in various traffic flow states [6–8].

In this study, the traffic flow is separated into six states by level of service (LOS). The main purpose is to identify the relationship between the LOS and different collision types and severities, and explore how contributing factors affect collision risks for different types and severities. The collision-related data were collected on the I-880 freeway of California in the United States from 2006 to 2011. The Bayesian conditional logit models have been established to analyze the statistical relationship between the collision probability of different collision patterns and LOSs. The five-stage Bayesian random parameter sequential logit model was established to quantify the effects of various variables on the collision probability of different types and severities. This research can help traffic management personnel better understand which LOS is more dangerous for different collision types and severities and realize the contributing factors of different collision types and severities in different LOSs. The results can be applied to reduce the collision probability of different types and severities in different LOSs.

## 2. Literature Review

Although most studies have explored the collision mechanism without considering the traffic flow states [1–5], some researchers analyzed the collision types and severities in different traffic flow states. In early studies, Golob et al. separated the traffic flow into various states. The researchers indicated that the traffic flow states with high densities could increase the probability of property damage only and multi-vehicle collisions, while the traffic flow states with low densities could increase the probability of single-vehicle and injury collisions [6]. Subsequently, Golob et al. separated the traffic flow into eight states and analyzed the relationship between collision types and traffic states. The results indicated that there is a significant difference in collision characteristics associated with various traffic states. However, there is insufficient qualitative analysis of the contributing factors in various traffic states [7].

Recently, Li et al. found that the speed-related variables can significantly affect the collision probability of different traffic flow states on a freeway. According to the speeds upstream and downstream of a crash, the traffic flow is separated into four states: back of queue, congested traffic, front of queue, and free flow. The results showed that the variation of speeds could increase the probability of a collision in free flow conditions, while the coefficient of speed variation could increase the probability of a collision in back of queue and congested traffic [8]. Subsequently, Li et al. found that there was a significant relationship between rear end collisions and the magnitude of lengthwise traffic variations, while sideswipe collisions were significantly related to the traffic variation between adjacent lanes on a freeway [9]. Wang et al. analyzed the short-term variation and spatial–temporal characteristics of traffic flow by sideswipe collisions. The results implied that the occurance of sideswipe collisions was significantly related to occupancy, average flow, and speed variance [10]. Kwak et al. defined the traffic flow states by uncongested and congested conditions. The results indicated that there was a significant difference of collision probability by different traffic states and road types [11]. Xu et al. applied four traffic states defined by four-phase traffic theory. The preliminary analysis showed that collision probability, as well as collision severities and types, were significantly affected by traffic flow states. Nonlinear canonical correlation analysis was applied to analyze the collision mechanism. The results showed that the contributing factors leading to the occurance of collisions were significantly different for varying traffic flow states [12]. Xu et al. developed collision probability models to explore the relationship between the probability of collisions and various traffic states separated by the three-phase traffic theory. The study implied that some transitional states were more dangerous than free flow, such as the transitional state from synchronized flow to free flow and the transitional state from wide moving jams to synchronized flow [13].

Numerous studies have focused on how traffic flow operates in different traffic states. The evolution of traffic dynamics on freeways is complex, and the formation of various traffic flow states is influenced by a set of factors. Therefore, traffic flow is classified into different states, typically based on traffic flow characteristics such as speed, flow rate, and density. Hall et al. separated the traffic flow into three states [14]. Wu separated the traffic flow into four states [15]. Kerner divided the traffic flow into three phases [16]. In this study, LOS is applied to separate the traffic states, which is one of the common methods for identifying traffic states. In LOS theory, traffic flow is separated into six states, according to density [17].

## 3. Data Sources

Crash data, environment data, geometric design data, and traffic flow data were collected from the I-880 freeway in California, United States between 2006 and 2011. The freeway is 34 miles in length and located between the cities of Oakland and San Jose. There are 119 loop detector stations and three weather stations along the freeway.

A total of 9919 collisions were reported and used for further data analysis. For every collision, to avoid the uncertainty of occurrence time, the collision-related data were collected from 5 min to 10 min before the occurrence time of the reported collision. In previous studies, this method has been proven to be effective [4,13]. Previous studies suggested that the statistical power is negligible by using a control-to-case ratio beyond 4:1 [13]. Thus, a control-to-case ratio of 4:1 was used in this study. For each collision case, the authors randomly selected four paired observations of the non-collision traffic data on the basis of three matching factors, including the time, the location, and the weather [13]. For example, collision No. 67 occurred at post-mile 3.95 at 15:00 on 9 November 2009. Traffic data taken at the nearest detector station from 2:50 p.m. to 2:55 p.m. on 9 November 2009 were included in the collision cases as an observation. Then, the paired collision-free traffic data taken at the same loop detector station during the same period on four randomly selected collision-free days in the same weather conditions were used as four observations in the non-collision cases. In this study, the severity of collision was divided into injury or fatal collisions and property damage only (PDO) collisions.

As shown in Table 1, the boundary values of density at different LOSs are presented. In this study, according to the LOS on the freeway, the traffic flow states were divided into six states. In addition, the statistical results in Table 2 show that the number of different collision types and severities in various LOSs are quite different.

**Table 1.** Boundary values of density at different levels of service (LOSs).

| LOS | Boundary Value of Density (Vehicle/km/Lane) |
|-----|---------------------------------------------|
| LOS A | ≤18 |
| LOS B | 18–29 |
| LOS C | 29–42 |
| LOS D | 42–56 |
| LOS E | 56–72 |
| LOS F | >72 |

**Table 2.** The number of collisions by different types and severities.

| LOS | Hit Object Collision | Sideswipe Collision | Rear end Collision | Injury Collision | Total |
|-----|---------------------|---------------------|--------------------|------------------|-------|
| LOS A | 1364 | 1888 | 4634 | 2386 | 8326 |
| LOS B | 54 | 139 | 593 | 199 | 811 |
| LOS C | 31 | 82 | 381 | 130 | 505 |
| LOS D | 5 | 28 | 134 | 43 | 169 |
| LOS E | 6 | 6 | 34 | 13 | 47 |
| LOS F | 6 | 13 | 37 | 19 | 61 |
| Total | 1466 | 2156 | 5813 | 2790 | 9919 |

## 4. Methods

In this study, the Bayesian conditional logit model was built to analyze the relative safety performance of different collision types and severities without considering other traffic-related factors in different LOSs. A five-stage Bayesian random parameter sequential logit model was applied to quantify the effects of various variables on the collision probability of different types and severities.

### 4.1. Bayesian Conditional Logit Model

In previous studies, the conditional logit model has already been used to analyze the safety performance of different traffic states [18,19]. The calculation method has been written as follows:

$$y_{ijk} \sim \text{Bernoulli}(p_{ijk}) \tag{1}$$

$$P(y_{itk}) = \frac{1}{\left\{1 + \exp\left[-\alpha_i + \sum_{k=1}^{K} \beta_k x_{ijk}\right]\right\}} \tag{2}$$

where $x_{ijk}$ is the $k$th unmatched factor for the $j$th sample or control in the ith matched sample. Therefore, $X = \{x_{ijk}\}$ consists of all samples, and all matched samples are controlled. The value of i is from 1 to I. The value of j is from 1 to $J$. The value of k is from 1 to $K$. $I$ denotes the number of matched samples; $J$ represents the number of controls in every matched sample; and $K$ represents the number of contributing factors. $\alpha_i$ is the effect of matching factors on the probability of collision occurance for each matched sample; $\beta_k$ represents the estimated value of contributing factors; and $x_k$ is the unmatched contributing factors.

To account for the selection bias introduced by the matched case–control design, a conditional likelihood needed to be developed. The conditional probability that the first vector of the explanatory variables $x_{i0}$ in the $i$th matched set corresponds to the case, conditional on $x_{i0}, x_{i1}, \ldots, x_{iJ}$ being the vectors of explanatory variables in the $i$th matched set, is given as

$$P_i^c = \frac{\exp\left(\sum_{k=1}^{K} \beta_k x_{i0k}\right)}{\exp\left(\sum_{k=1}^{K} \beta_k x_{i0k}\right) + \sum_{j=1}^{J} \exp\left(\sum_{k=1}^{K} \beta_k x_{ijk}\right)} \tag{3}$$

Thus, the likelihood function of the conditional logit can be written as [15]

$$f(Y|\beta) = \prod_{i=1}^{I} f(y_{i0} = 1|\beta) = \prod_{i=1}^{I} P_i^c$$
$$= \exp\left\{\sum_{i=1}^{I}\sum_{k=1}^{K} (\beta_k x_{i0k}) - \sum_{i=1}^{I} \log\left[\sum_{j=0}^{J} \exp\left(\sum_{k=1}^{K} \beta_k x_{ijk}\right)\right]\right\} \tag{4}$$

The Bayesian inference method has been applied for this model using Markov Chain Monte Carlo (MCMC) methods, because there is a significant advantage of this method in that all parameters in the model have a prior distribution. The posterior distribution of parameters has been expressed as

$$f(\beta|Y) = \frac{f(Y,\beta)}{f(Y)} = \frac{f(Y|\beta)\pi(\beta)}{\int f(Y,\beta)d\beta} \propto f(Y|\beta)\pi(\beta) \tag{5}$$

where f ($\beta$ |Y) is a posterior joint probability distribution (JPD) associate with parameter $\beta$, based on data set $Y$; f(Y, $\beta$) is a JPD associate with parameter $\beta$ and data set Y; f(Y|$\beta$) denotes the probability conditional associated with parameter $\beta$; and $\pi(\beta)$ is a prior distribution associated with parameter $\beta$. The non-informative prior distribution in this method has been written as

$$\beta \sim \text{Normal}(0_K, 10^6 I_K) \tag{6}$$

where $0_K$ represents a $K \times 1$ vector of zeros and $I_K$ represents a $K \times K$ matrix. Finally, the posterior JPD $f(\beta\,|Y)$ has been written as

$$f(\beta|Y) \propto f(Y|\beta)\pi(\beta) = \prod_{i=1}^{I} f(y_{i0} = 1|\beta) \times \prod_{k=1}^{K} N(\beta_k|\mu_k, \Sigma_k)$$
$$\propto \exp\left\{ \sum_{i=1}^{I}\sum_{k=1}^{K}(\beta_k x_{i0k}) - \sum_{i=1}^{N} \log\left[ \sum_{j=0}^{J} \exp\left( \sum_{k=1}^{K} \beta_k x_{ijk} \right) \right] - \frac{1}{2}\sum_{k=1}^{K} \frac{(\beta_k)^2}{10^6} \right\} \tag{7}$$

## 4.2. Bayesian Random Parameter Sequential Logit Model

In previous studies, the ordered logit model was one of the popular methods used to analyze collision severities. However, there are some limitations of this method for analyzing collision severities as follows:

1.  There is a hypothesis of this method that the parameter estimates of different collision types and severities are the same [20,21]. However, compared to the ordered logit model, the sequential logit model can explain the difference of various contributing factors across different collision types and severities [20,21].
2.  In addition, the sequential logit model explains the correlation of collision probability between different collision types and severities [22,23]. The expressions of collision probability by different collision types and severities have been calculated by Equations (8) through (13), respectively.
3.  Moreover, collisions were affected by various traffic-related factors [24–26]. Thus, there is an unobserved heterogeneity in the sequential logit model [27–29]. The contributing factors in this study can not explain all of the variance in collision types and severities. The unobserved heterogeneity in models can result in inconsistent and biased estimation [30–32]. To overcome the limitation of unobserved heterogeneity in the sequential logit model, random parameters were applied in this study.

Therefore, the five-stage Bayesian random parameters sequential logit model was applied to calculate the collision probability of different severities and types. As shown in Figure 1, four Bayesian random parameters binary logit models were built from Stage 1 to Stage 2. Subsequently, three Bayesian random parameters binary logit models were built at the Stage 5. These Bayesian random parameters binary logit models formed the whole five-stage Bayesian random parameters sequential logit model.

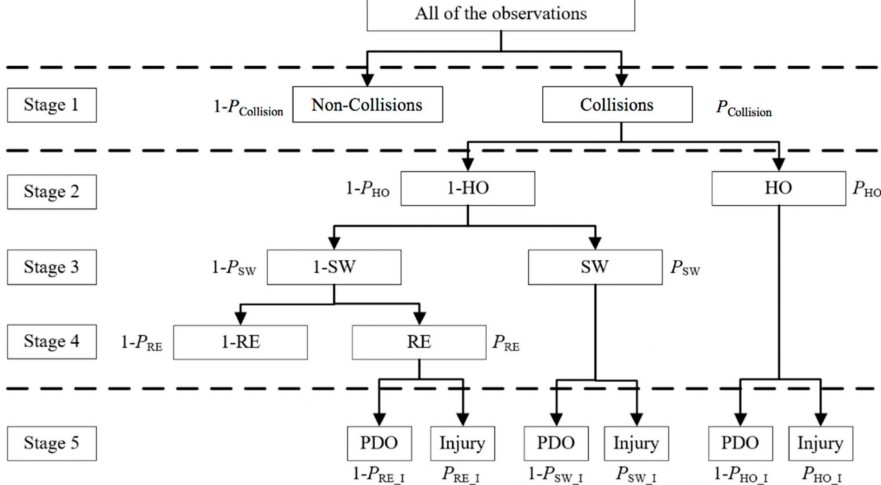

**Figure 1.** The framework of the five-stage sequential logit model.

Specifically, the Stage 1 model calculated the collision probability ($P_{Collision}$) without considering different collision types and severities. The Stage 2 model calculated the probability of a hit object collision ($P_{Collision} \times P_{HO}$) without considering collision severities. The Stage 3 model calculated the probability of a sideswipe collision ($P_{Collision} \times (1-P_{HO}) \times P_{SW}$) without considering collision severities. The Stage 4 model calculated the probability of a rear end collision ($P_{Collision} \times (1-P_{HO}) \times (1-P_{SW}) \times P_{RE}$) without considering collision severities. Finally, three collision severity models were established at Stage 5. The injury probability of a hit object collision is $P_{HO\_I}$, the injury probability of a sideswipe collision is $P_{SW\_I}$, and the injury probability of a rear end collision is $P_{RE\_I}$. Finally, the absolute probability of collision by different types and severities have been given as follows:

$$
\begin{aligned}
&\text{P(Hit object collision with injury)} = \text{P(Collision)} \\
&\times \text{P(Hit object collision|Collision)} \times \text{P(injury collision|Hit object collision)} = P_{Collision} \times P_{HO} \times P_{HO\_I}
\end{aligned}
\tag{8}
$$

$$
\begin{aligned}
&\text{P(Hit object collision without injury)} = \text{P(Collision)} \times \text{P(Hit object collision | Collision)} \\
&\times \text{P(PDO collision | Hit object collision)} = P_{Collision} \times P_{HO} \times (1 - P_{HO\_I})
\end{aligned}
\tag{9}
$$

$$
\begin{aligned}
&\text{P(Sideswipe collision with injury)} = \text{P(Collision)} \times \text{P(Non-Hit object collision} \\
&\text{| Collision)} \times \text{P(Sideswipe collision | Non-Hit object collision)} \times \text{P(injury collision} \\
&\text{| Sideswipe collision)} = P_{Collision} \times (1 - P_{HO}) \times P_{SW} \times P_{SW\_I}
\end{aligned}
\tag{10}
$$

$$
\begin{aligned}
&\text{P(Sideswipe collision without injury)} = \text{P(Collision)} \times \text{P(Non-Hit object} \\
&\text{collision|Collision)} \times \text{P(Sideswipe collision|Non-Hit object collision)} \times \text{P(PDO} \\
&\text{collision|Sideswipe collision)} = P_{Collision} \times (1 - P_{HO}) \times P_{SW} \times (1 - P_{SW\_I})
\end{aligned}
\tag{11}
$$

$$
\begin{aligned}
&\text{P(Rear end collision with injury)} = \text{P(Collision)} \times \text{P(Non-Hit object collision|Collision)} \\
&\times \text{P(Non-Sideswipe collision|Non-Hit object collision)} \times \text{P(Rear end collision|Non-} \\
&\text{Sideswipe collision)} \times \text{P(injury collision|Rear end collision)} = P_{Collision} \times (1 - P_{HO}) \times (1 - P_{SW}) \\
&\times P_{RE} \times P_{RE\_I}
\end{aligned}
\tag{12}
$$

$$
\begin{aligned}
&\text{P(Rear end collision without injury)} = \text{P(Collision)} \times \text{P(Non-Hit object} \\
&\text{collision|Collision)} \times \text{P(Non-Sideswipe collision|Non-Hit object collision)} \times \text{P(Rear end} \\
&\text{collision|Non-Sideswipe collision)} \times \text{P(PDO collision|Rear end collision)} = P_{Collision} \times \\
&(1 - P_{HO}) \times (1 - P_{SW}) \times P_{RE} \times (1 - P_{RE\_I})
\end{aligned}
\tag{13}
$$

The specification of the basic Bayesian random parameters binary logit model has already been introduced in previous studies [33,34].

## 5. Results and Discussion

To analyze the relative safety performance of various collision types and severities between different LOSs, the Bayesian conditional logit model was used in Section 5.1. In the Bayesian conditional logit models, only LOSs were regarded as candidate variables without considering other variables.

To quantify the effects of various variables on the collision probability of different collision types and severities in various LOSs, the five-stage Bayesian random parameter sequential logit model was applied in Section 5.2. In addition to LOS variables, five other candidate variables were also considered in the five-stage Bayesian random parameter sequential logit model.

### 5.1. Safety Performance of LOS by Different Collision Types and Severities

According to the LOS on the freeway, the Bayesian conditional logit model was used to analyze the relative safety performance of different collision types and severities without considering other traffic-related factors in different LOSs. There are five indicator variables in this model, including LOS B, LOS C, LOS D, LOS E, and LOS F. LOS A was considered as the reference level. Therefore, the purpose of this section is to explore the relative safety performance between LOS A and other LOSs. Other traffic

flow variables were not included in this model, because LOSs were highly correlated with the traffic flow variables [34].

The process of MCMC chains for this model was composed of a total of 10,000 iterations, 4000 burn-in iterations, and three parallel MCMC chains for Bayesian inference [30,31]. The results of the Bayesian conditional logit models are shown in Table 3. The results show that LOSs significantly affect the collision probability for different types and severities. The 95% credible interval for each parameter in Table 3 indicates that the LOSs significantly affect the collision probability of different types and severities. The odds ratio for each variable was used to quantify the safety performance of each LOS.

**Table 3.** The results of Bayesian conditional logit models.

| Variables | Mean | MC Error | 2.50% | 97.50% | Odds Ratio |
|---|---|---|---|---|---|
| **Hit Object Collision** | | | | | |
| LOS B | 1.057 | 0.202 | 0.656 | 1.454 | 2.878 |
| LOS C | 1.463 | 0.262 | 0.937 | 1.956 | 4.319 |
| LOS D | 1.040 | 0.668 | −0.373 | 2.256 | 2.829 |
| LOS E | 1.183 | 0.604 | −0.043 | 2.331 | 3.264 |
| LOS F | 1.023 | 1.121 | −1.142 | 3.270 | 2.782 |
| LOS A * | | | | | |
| **Sideswipe Collision** | | | | | |
| LOS B | 1.141 | 0.126 | 0.898 | 1.379 | 3.130 |
| LOS C | 1.470 | 0.157 | 1.174 | 1.784 | 4.349 |
| LOS D | 1.985 | 0.287 | 1.443 | 2.547 | 7.279 |
| LOS E | 0.484 | 0.655 | −0.908 | 1.668 | 1.623 |
| LOS F | 0.737 | 0.821 | −0.983 | 2.290 | 2.090 |
| LOS A * | | | | | |
| **Rear end Collision** | | | | | |
| LOS B | 1.385 | 0.065 | 1.264 | 1.515 | 3.995 |
| LOS C | 1.797 | 0.085 | 1.628 | 1.963 | 6.032 |
| LOS D | 1.957 | 0.136 | 1.69 | 2.223 | 7.078 |
| LOS E | 1.623 | 0.242 | 1.153 | 2.089 | 5.053 |
| LOS F | 1.757 | 0.353 | 1.078 | 2.448 | 5.795 |
| LOS A * | | | | | |
| **Injury Collision** | | | | | |
| LOS B | 1.345 | 0.102 | 1.148 | 1.541 | 3.838 |
| LOS C | 1.594 | 0.132 | 1.329 | 1.857 | 4.923 |
| LOS D | 1.699 | 0.216 | 1.273 | 2.118 | 5.468 |
| LOS E | 1.251 | 0.372 | 0.497 | 1.977 | 3.494 |
| LOS F | 1.808 | 0.527 | 0.78 | 2.828 | 6.098 |
| LOS A * | | | | | |

* denotes the reference level.

Specifically, as shown in Table 3 for hit object collisions, the results suggest that the odds ratios of LOS B and LOS C were significantly greater than LOS A, and the odds ratios of LOS D, LOS E, and LOS F were not significantly greater than LOS A. Accordingly, LOS A was the safest traffic state according to the lowest hit object collision probability. However, LOS C had the highest hit object collision likelihood and was 3.319 times higher than LOS A. The hit object collision probability of LOS B was 1.878 times higher than LOS A, but lower than LOS C. In previous studies, the results indicated that a hit object collision was more likely to occur in traffic flow states with low density [6,7]. In this study, LOS C had a higher density than LOS A and LOS B. Thus, LOS C was the most dangerous for hit object collisions in all LOSs. The analysis of LOS C can also be applied to the results of LOS B.

The results of sideswipe collisions are shown in Table 3. Three odds ratios were significantly greater than LOS A, including LOS B, LOS C, and LOS D. Two odds ratios were not significantly greater than LOS A, including LOS E and LOS F. The highest sideswipe collision probability was for LOS D,

followed by LOS C and LOS B. LOS A had the lowest sideswipe collision probability. Specifically, the sideswipe collision probability of LOS B was 2.130 times higher than LOS A, the sideswipe collision probability of LOS C was 3.349 times higher than LOS A, and the sideswipe collision probability of LOS D was 6.279 times higher than LOS A. LOS D had the highest density, followed by LOS C, LOS B, and LOS A. There were more and more lane-changing behaviors in traffic flow with the density increasing. More lane-changing behaviors can increase the risk of a sideswipe collision [16]. Thus, the highest sideswipe collision probability was for LOS D, followed by LOS C, LOS B, and LOS A.

In the Bayesian conditional logit model for rear end collisions, the results showed that there were some significant differences in different LOSs. All other LOSs were more dangerous than LOS A for rear end collisions. The highest rear end collision probability was for LOS D, followed by LOS C, LOS F, LOS E, and LOS B. Specifically, the rear end collision probability of LOS B was 2.995 times higher than LOS A, the rear end collision probability of LOS C was 5.032 times higher than LOS A, the rear end collision probability of LOS D was 6.078 times higher than LOS A, the rear end collision probability of LOS E was 4.053 times higher than LOS A, and the rear end collision probability of LOS F was 4.795 times higher than LOS A. The reason why LOS D had the highest rear end collision probability is similar to the sideswipe collisions above. In addition, although LOS E and LOS F had higher densities than LOS D, there was less space in LOS E and LOS F for vehicles to change lanes. Thus, LOS D had the highest rear end collision probability. Although LOS C was still in free flow, some transitional states from free flow to congested flow started to emerge with sudden reductions in speed. This is the reason why LOS C had the second-highest rear end collision probability.

As shown in Table 3, the estimation results of the Bayesian conditional logit model for injury collisions show that the LOS significantly affected the probability of injury collision occurrences. LOS F had the highest injury collision probability, followed by LOS D, LOS C, LOS B, and LOS E. Specifically, the injury collision probability of LOS B was 2.838 times higher than LOS A, the injury collision probability of LOS C was 3.923 times higher than LOS A, the injury collision probability of LOS D was 4.468 times higher than LOS A, the injury collision probability of LOS E was 2.494 times higher than LOS A, and the injury collision probability of LOS F was 5.098 times higher than LOS A. It has been proven that a higher density can lead to injury collisions [6,7]. Thus, LOS F had the highest injury collision probability. In LOS D and LOS C, more transitional states from free flow to congested flow started to emerge with sudden reductions in speed. Due to LOS D having a higher density than LOS C, LOS D had the second-highest injury collision probability, followed by LOS C.

*5.2. The Sequential Logit Model for Collision Types and Severities*

The five-stage Bayesian random parameter sequential logit model was established to quantify the relationship between LOS and collision probability by different types and severities. Specifically, Section 5.2.1 was used to explore the relationship between LOS and collision types. In this section, Stage 1 is used to predict the collision likelihood. Stages 2–4 are used to predict the hit object collision probability, sideswipe collision probability, and rear end collision probability, respectively. Section 5.2.2 was used to explore the relationship between LOS and collision severities. In this section, Stage 5 is used to predict the injury probability for different collision types. To avoid the correlation between traffic flow variables and LOS, only LOS was considered in the models. In addition to LOS, as shown in Table 4, five other candidate variables including visibility, road surface, weather, ramp, and number of lanes were also taken into consideration at every stage. The simulation method of this section is similar to that of Section 4.1.

**Table 4.** Candidate variables.

| Candidate Variables | Explanation |
|---|---|
| Vi | Visibility (mile) |
| We | 1 = worse weather conditions; 0 = normal weather conditions; |
| Rs | 1 = worse road surface; 0 = normal road surface |
| Ra | 1 = ramp segment; 0 = non-ramp segment |
| Nl | Number of lanes |
| LOS A | 1 = LOS A; 0 = otherwise |
| LOS B | 1 = LOS B; 0 = otherwise |
| LOS C | 1 = LOS C; 0 = otherwise |
| LOS D | 1 = LOS D; 0 = otherwise |
| LOS E | 1 = LOS E; 0 = otherwise |
| LOS F | 1 = LOS F; 0 = otherwise |

### 5.2.1. Sequential Model for Collision Types

Table 5 presents the results of the collision probability for different types, from Stage 1 to Stage 4. As shown at Stage 1, low visibility significantly increases collision probability. LOS A has a random negative coefficient, indicating that the collision probability decreases in LOS A. In previous studies, it has been proven that free flow has the best safety performance [15] because the driver has sufficient time to adopt emergency measures in LOS A, with less flow and more space. LOS C has a positive correlation effect on the collision probability. Although vehicles are still in free flow in LOS C, the space between vehicles becomes smaller than in LOS A and LOS B. In previous studies, it has been demonstrated that more drivers will take advantage of higher speeds under uncongested conditions [35,36]. Thus, higher speed and smaller space can lead to less response time for drivers to take emergency measures.

**Table 5.** Results of sequential model from Stage 1 to Stage 4.

| Variables | Mean | MC Error | 2.50% | Median | 97.50% |
|---|---|---|---|---|---|
| **Stage 1** | | | | | |
| Vi | −0.111 | 0.014 | −0.136 | −0.126 | −0.001 |
| LOS A | −0.017 | 0.003 | −0.025 | −0.018 | 0.000 |
| LOS C | 0.049 | 0.009 | 0.013 | 0.051 | 0.082 |
| **Stage 2** | | | | | |
| Nl | −0.146 | 0.008 | −0.196 | −0.150 | −0.073 |
| Vi | −0.153 | 0.002 | −0.163 | −0.155 | −0.141 |
| Rs | 0.264 | 0.023 | 0.008 | 0.287 | 0.458 |
| LOS B | −0.211 | 0.020 | −0.414 | −0.169 | −0.049 |
| LOS C | −0.279 | 0.023 | −0.390 | −0.335 | −0.016 |
| LOS D | −0.622 | 0.061 | −1.078 | −0.539 | −0.054 |
| **Stage 3** | | | | | |
| Nl | −0.033 | 0.002 | −0.061 | −0.031 | −0.017 |
| Ra | −0.255 | 0.041 | −0.655 | −0.135 | −0.001 |
| Vi | −0.106 | 0.002 | −0.119 | −0.107 | −0.087 |
| LOS A | −0.045 | 0.002 | −0.059 | −0.046 | −0.015 |
| LOS B | −0.137 | 0.015 | −0.241 | −0.164 | −0.033 |
| LOS C | −0.272 | 0.019 | −0.398 | −0.301 | −0.019 |
| LOS D | −0.469 | 0.056 | −0.908 | −0.614 | −0.010 |
| **Stage 4** | | | | | |
| Nl | 0.170 | 0.005 | 0.097 | 0.181 | 0.198 |
| Vi | 0.196 | 0.003 | 0.151 | 0.199 | 0.208 |
| Rs | 0.158 | 0.013 | 0.029 | 0.165 | 0.268 |
| LOS A | 0.030 | 0.003 | 0.005 | 0.029 | 0.059 |
| LOS C | 0.201 | 0.022 | 0.086 | 0.136 | 0.372 |
| LOS D | 0.126 | 0.011 | 0.011 | 0.109 | 0.261 |

As shown in Table 5, the hit object collision probability was calculated for Stage 2. This model has six significant variables, as shown in Table 5, including number of lanes, visibility, road surface, LOS B, LOS C, and LOS D. The results indicated that the hit object collision probability increases with less lanes, low visibility, and a worse road surface. Specifically for these reasons, less lanes can lead to less space between vehicles. Low visibility can result in less response time for drivers to take emergency measures. Vehicles need longer braking distances in worse road surface conditions. In addition, the hit object collision probability decreases in LOS B, LOS C, and LOS D.

The Stage 3 model was established to predict the likelihood of a sideswipe collision. This model has seven significant variables, as shown in Table 5, including number of lanes, visibility, ramp, LOS A, LOS B, LOS C, and LOS D. Number of lanes, visibility and ramp have random negative coefficients, indicating that the sideswipe collision probability increases with less lanes, low visibility, and non-ramp conditions. In LOS A, LOS B, LOS C, and LOS D situations, the sideswipe collision probability decreases. The results indicated that low occupancy can decrease the sideswipe collision probability because there is more space for drivers to take emergency measures in low occupancy conditions.

The Stage 4 model was established to predict rear end collision likelihood. This model has six significant variables, as shown in Table 5, including number of lanes, visibility, road surface, LOS A, LOS C, and LOS D. More lanes, high visibility, and a worse road surface have positive effects on rear end collision probability. For LOS A, LOS C, and LOS D, the rear end collision probability increases.

### 5.2.2. Sequential Model for Collision Severities by Different Types

Table 6 presents the results of the probability of collision severities by different types at Stage 5. It was found that there are significant differences in the contributing factors of the estimation results.

**Table 6.** Results of sequential model for Stage 5.

| Variables | Mean | MC Error | 2.50% | Median | 97.50% |
| --- | --- | --- | --- | --- | --- |
| **Hit Object Collision** | | | | | |
| Nl | −0.048 | 0.008 | −0.073 | −0.057 | −0.001 |
| Ra | −0.062 | 0.013 | −0.130 | −0.059 | −0.006 |
| We | −0.095 | 0.012 | −0.129 | −0.110 | −0.029 |
| Vi | −0.040 | 0.003 | −0.052 | −0.041 | −0.016 |
| **Sideswipe Collision** | | | | | |
| Nl | −0.055 | 0.006 | −0.068 | −0.059 | −0.011 |
| Ra | −0.079 | 0.014 | −0.168 | −0.072 | −0.004 |
| Vi | −0.100 | 0.010 | −0.120 | −0.111 | −0.020 |
| LOS D | −0.367 | 0.051 | −0.479 | −0.431 | −0.050 |
| **Rear end Collision** | | | | | |
| Nl | −0.073 | 0.004 | −0.102 | −0.075 | −0.036 |
| We | −0.074 | 0.008 | −0.140 | −0.068 | −0.002 |
| Vi | −0.049 | 0.002 | −0.069 | −0.050 | −0.034 |
| Rs | −0.115 | 0.023 | −0.348 | −0.067 | −0.017 |
| LOS A | −0.021 | 0.002 | −0.036 | −0.022 | −0.003 |

For hit object collision, the results indicate that four variables can significantly affect the severity of hit object collisions. All of the significant variables have negative effects on the injury probability of hit object collisions, specifically few number of lanes, non-ramp segments, normal weather, and low visibility can increase the injury probability of hit object collisions. The results indicate that higher speeds in non-ramp segments, normal weather conditions, fewer number of lanes, and less response time in low-visibility conditions can increase the injury probability of hit object collisions. In addition, it was found that LOS has no effects on the severity of hit object collisions.

For sideswipe collisions, the results imply that four variables can significantly affect the severity of sideswipe collisions. Specifically, fewer number of lanes, non-ramp segments, and low visibility can

increase the injury probability of sideswipe collisions. The results are similar to the hit object collisions above. Moreover, a PDO sideswipe collision is more likely to occur in LOS D.

For rear end collisions, the results indicate that five variables can significantly affect the severity of rear end collisions. Specifically, few number of lanes, non-ramp segments, low visibility, and normal road surfaces can increase the injury probability of rear end collisions. Furthermore, a PDO rear end collision is more likely to occur in LOS A.

## 6. Conclusions

In this study, the main purpose was to identify the relationship between LOS and different collision types and severities, and explore how contributing factors affect collision risks for different types and severities. The collision-related data were obtained from the I-880 freeway, which is located in California, United States. The time interval was from 2006 to 2011. The Bayesian conditional logit model was built to analyze the relative safety performance of different collision types and severities without considering other traffic-related factors in different LOSs. A five-stage Bayesian random parameter sequential logit model was applied to quantify the effects of various variables of the collision probability of different types and severities.

Specifically, as shown in Figure 2, the results of the Bayesian conditional logit models in Table 3 indicate that LOS A is the safest traffic state for different collision types and severities. LOS C has the worst safety performance associated with hit object collisions, and the hit object collision probability in LOS C is 3.319 times higher than the one in LOS A. LOS D has the worst safety performance associated with sideswipe collisions and rear end collisions, the sideswipe collision probability and the rear end collision probability in LOS D is 6.279 and 6.078 times higher than the one in LOS A. LOS F has the worst safety performance associated with injury collisions, and the injury collision probability in LOS F is 5.098 times higher than the one in LOS A, because an injury collision is more likely to occur in traffic flow states with high occupancy [6,7].

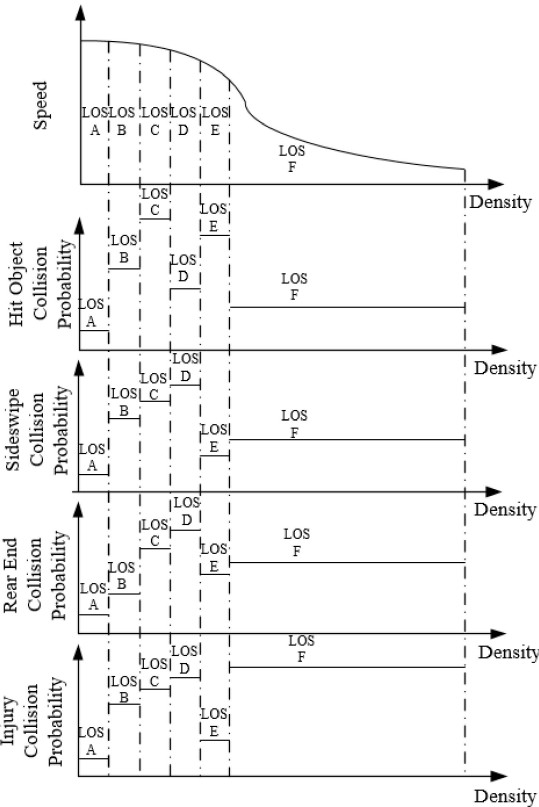

**Figure 2.** The collision probability of different types and severities in various LOSs.

The results of the sequential logit model showed that weather variables, road variables, and LOS had significant effects on different collision types and severities. It was found that fewer lanes and low visibility can both increase the injury probability of hit object collisions, sideswipe collisions, and rear end collisions. Ramp segments could decrease the injury probability of hit object collisions and sideswipe collisions. Normal weather conditions could increase the injury probability of hit object collisions and rear end collisions. Normal road surfaces could increase the injury probability of rear end collisions.

This research can help transportation professionals better understand which LOS is more dangerous for different collision types and severities, and realize the contributing factors of different collision types and severities in different LOSs. The results can be applied to reduce the collision probability of different types and severities in different LOSs.

However, there are still some issues that need to be studied in the future. Firstly, more divided methods of traffic flow states should be adapted, such as three-phase theory. Second, more traffic variables should be used in the models, such as driver behavior and geometric design. Finally, the transferability of the models in this study still needs to be verified in the future.

**Author Contributions:** Conceptualization, B.Y., Y.W.; methodology, B.Y.; software, J.B.; validation, B.Y., J.B., W.Z.; formal analysis, B.Y.; investigation, W.Z.; resources, W.Z.; data curation, Y.W.; writing—original draft preparation, B.Y.; writing—review and editing, B.Y.; visualization, J.B.; supervision, B.Y.; project administration, W.Z.; funding acquisition, Y.W. All authors have read and agreed to the published version of the manuscript.

**Funding:** This research was sponsored by the Projects of the National Natural Science Foundation of China (71701046; 51878236), the Foundation for Jiangsu key Laboratory of Traffic and Transportation Security (TTS2020-03), and the Talent Research Start-Up Fund of Nanjing University of Aeronautics and Astronautics under grant (1007-YAH20100).

**Conflicts of Interest:** The authors declare no conflict of interest.

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
