# Peer review of "Modeling Collision Probability on Freeway: Accounting for Different Types and Severities in Various LOS"

_sustainability, doi:10.3390/su12187386_

Round 1

Reviewer 1 Report

Modeling Collision Probability on Freeway: Accounting for Different Types and Severities in Various LOS

The purpose of this paper is to identify how the level of service affects collision types and severities. The paper has some interesting relationships outlined and would provide good information for practitioners and researchers. There are several changes that should be made to the paper though before it is considered for publication.

The following comments are provided listed by line number:

Line 2 – “Collision” is misspelled in the title.

Line 18 – The word “object” should be “objective.” This is the case in several instances in the paper.

Line 20 – Change “…been establish…” to “…been established…”

Line 21 – The word “analyzing” should be “analyze.”

Line 22 – In this line and throughout the entire paper when referring to level of service, use the acronym LOS in all cases (as opposed to Los).

Line 23 – Is the crash type really “hit object”? This sounds awkward.

Line 28 – The word “significantly” should be “significant” here and in many other places in the document. This line should read “…number of lanes have significant effects on…”

Line 44 – Change this sentence to read “…have investigated the collision types under different traffic conditions.”

Line 47 – The word “sates” should be “states.”

Line 55 – The phrase “…the speeds at upstream and downstream, the traffic flow is…” is confusing to me. What do you mean by upstream and downstream? Upstream and downstream of what?

Line 60, 66 – Change “significantly” to “significant.”

Line 67 – It is recommended that the use of numbers be done consistently. The general rule is to spell out one through nine and use numbers for 10 and above. The exception to that rule is to always spell out a unit of measurement. I would not necessarily call traffic states as a unit of measurement so I would recommend “four different traffic states defined by four-phase traffic theory.”

Line 69 – Why are there areas throughout the paper that are highlighted?

Line 86 – What is S, F, and J? These need to be defined.

Line 90 – Define level of service here as LOS and then use LOS throughout the rest of the paper (not Los).

Line 94 – Remove “the” from this line “…LOS affects collision types and severities.”

Line 98 – Rewrite as “…that focus on preventing specific types of collisions.”

Introduction in General – The introduction has some good information and background. There is a statement that says, “The present study aims at identifying how LOS affects collision types and severities.” It is recommended that this more directly state the “purpose” as a purpose rather than an aim. The next line then says “The research results could help transportation professionals…” This is not a very direct statement on the impact of the research. Be more direct in stating the purpose and need and making it very clear what the contribution of the research is. It was unclear in the paper what the overall contribution is to the profession.

Line 100 – Provide a better transition between sections.

Line 102 – Use “three” in place of “3.”

Line 110 – No “the” needed before “Caltrans.”

Line 112 – Rewrite as “The injury collisions in this study are injury…and the non-injury collision is a property damage only (PDO) collision.” Define PDO here as an acronym and then use just the acronym later on.

Line 114 – Use LOS, not Los. This is applicable throughout the remainder of the paper as Los is used quite often (more than LOS).

Line 115 – Once an acronym is defined (in this case LOS), use the acronym consistently throughout the remainder of the document. This is the case in a couple of locations in this paragraph.

Line 117 – Refer to the tables by number. It is also best to refer to the table before the table appears in the document. In that case, this paragraph would come before the table, rather than after the table.

Line 118 – Again, refer to the table by number. This paragraph overall (and several others in the document) is quite choppy. It jumps around quite a bit. Look at ways to make the writing overall flow better.

Line 127 – The word “mode” here seems like it should be “model.” This is the case throughout this paragraph.

Line 134 – How does Figure 1 show the absolute probability of collision by different types and severities can be calculated by all these equations? It is unclear to me how this “shows” the equations. You need a better connection between the figure and the equations along with a better discussion on both to help the reader understand the process better.

Line 141 – Remove the word “metric” as it seems that “geometric” is sufficient. What other variables could be included? Rather than use “etc.” include an exhaustive list if possible.

Line 169 – MCMC is already defined as an acronym.

Line 177 – Change “assume” to “assumed.”

Line 181 – It seems that “mode” here should be “model”?

Line 183 – Be more specific on where the collision types and severities were calculated. Include equation numbers and clarify. Overall the discussion of the model is quite confusing.

Line 187 – References 23 through 31 are all lumped together as part of one sentence. That is a lot of references to tie into one sentence. More detail on this should be provided.

Line 189 – Should this refer to Figure 1? It does not make sense to reference Figure 2 at this point in the document.

Line 191/192 – In line 191 you refer to “three random parameters” and then in line 192 you say “These seven random parameters…” Is it three or seven? Or do these even go together? This needs to be clarified.

Line 217 – Instead of “with over lines…” you could refer to them as bars.

Line 221 – Provide an introductory paragraph to the subsections. Also, the word “od” should be “of” I believe. Be sure to proofread and spell-check.

Line 223 – Always use a capital letter for the LOS designations (in this case “E.”

Line 225 – Use the acronym LOS in place of “level of service” once it has been defined.

Line 227 – Add “the” after “…highly correlated with…”

Line 229 – Use LOS throughout the table. What is the baseline value for LOS A? It is unclear and the discussion compares everything with LOS A but it is never clarified what that value is.

Line 234 – Remove “the” at the end of this line.

Line 242 – The discussion provided for the “hit object collision” does not make sense. I see similar odds ratios for all LOS. How is LOS B and LOS C “significantly greater than LOS A”? LOS E has a higher odds ratio than LOS B so this doesn’t make sense. There is some confusion in this discussion.

Line 270 – Rewrite as “In addition to LOS, five other candidate variables including…were also taken into consideration at every stage.”

Line 274 – Provide a better transition between section 4.2 and the subsections that follow it.

Lines 276-280 – Which columns are you comparing here? The results do not match what is in the table. Later on, you always refer to the mean values as the comparison. The coefficients on the mean values are always negative so how can some probabilities increase and some decrease? The discussion does not match the results. This is also true in the stage 2 model discussion. Please review all discussions and update accordingly.

Line 298 – This section has a better discussion that matches what the tables show. One thought in this section is to only use the acronym PDO as it should have been defined several sections back.

Line 314 – The conclusions and discussion section should first clearly restate the purpose and need, identify how the purpose and need were addressed, and then provide a discussion of the results. There are several changes to make with the English in this section including changing “establish” to “established” in Line 316, “analyzing” to “analyze” in Line 317, “predict” to “predicted” in Line 320, and “predict” to “predicted” in Line 321. The conclusions need to also make it very clear what the contribution is. I do not walk away from this paper with a true understanding of the overall contribution that this research makes to the literature. This is a critical change that needs to be made.

Author Response

Response to Reviewer 1 Comments

Point 1: Line 2 – “Collision” is misspelled in the title.

Response 1: The spelling mistake has been revised in line 2.

Point 2: Line 18 – The word “object” should be “objective.” This is the case in several instances in the paper.

Response 2: The word has been revised in line 18.

Point 3: Line 20 – Change “…been establish…” to “…been established…”

Response 3: The mistake has been revised in line 20.

Point 4: Line 21 – The word “analyzing” should be “analyze.”

Response 4: The mistake has been revised in line 21.

Point 5: Line 22 – In this line and throughout the entire paper when referring to level of service, use the acronym LOS in all cases (as opposed to Los).

Response 5: The acronym Los has been replaced by LOS in all cases.

Point 6: Line 23 – Is the crash type really “hit object”? This sounds awkward.

Response 6: The hit object crash has been frequently used in previous studies. It can be inspected in references as followed:

Golob, T.; Recker, W.; Alvarez, V. Freeway Safety as a Function of Traffic Flow. Accident Analysis and Prevention. 2004, 36(6), 933-946.

Golob, T.; Recker, W. A Method for Relating Type of Crash to Traffic Flow Characteristics on Urban Freeways. Transportation Research Part A. 2004, 38(1), 53-80.

Point 7: Line 28 – The word “significantly” should be “significant” here and in many other places in the document. This line should read “…number of lanes have significant effects on…”

Response 7: The mistake has been revised in line 28. The same mistakes in other cases have also been revised.

Point 8: Line 44 – Change this sentence to read “…have investigated the collision types under different traffic conditions.”

Response 8: The sentence has been changed in line 44.

Point 9: Line 47 – The word “sates” should be “states.”

Response 9: The introduction of original version has been rewritten and separated into two section, including introduction and literature review. And the word “sates” has been revised in line 64.

Point 10: Line 55 – The phrase “…the speeds at upstream and downstream, the traffic flow is…” is confusing to me. What do you mean by upstream and downstream? Upstream and downstream of what?

Response 10: The speeds at upstream and downstream of crash has been shown in line 71.

Point 11: Line 60, 66 – Change “significantly” to “significant.”

Response 11: The mistake has been revised in line 76 and 82.

Point 12: Line 67 – It is recommended that the use of numbers be done consistently. The general rule is to spell out one through nine and use numbers for 10 and above. The exception to that rule is to always spell out a unit of measurement. I would not necessarily call traffic states as a unit of measurement so I would recommend “four different traffic states defined by four-phase traffic theory.”

Response 12: The proposals have been accepted in line 83 and all the other cases.

Point 13: Line 69 – Why are there areas throughout the paper that are highlighted?

Response 13: The areas throughout the paper that are highlighted are not in the original submission version. Maybe it was edited by editor.

Point 14: Line 86 – What is S, F, and J? These need to be defined.

Response 14: Line 90 – S, F, and J in sentence are replaced by synchronized flow, free flow and wide moving jams respectively.

Point 15: Line 90 – Define level of service here as LOS and then use LOS throughout the rest of the paper (not Los).

Response 15: The proposals have been accepted in all case.

Point 16: Line 94 – Remove “the” from this line “…LOS affects collision types and severities.”

Response 16: The introduction of original version has been rewritten and separated into two section, including introduction and literature review. And this mistake has no longer existed.

.

Point 17: Line 98 – Rewrite as “…that focus on preventing specific types of collisions.”

Response 17: The sentence has been rewritten in line 56.

Point 18: Introduction in General – The introduction has some good information and background. There is a statement that says, “The present study aims at identifying how LOS affects collision types and severities.” It is recommended that this more directly state the “purpose” as a purpose rather than an aim. The next line then says “The research results could help transportation professionals…” This is not a very direct statement on the impact of the research. Be more direct in stating the purpose and need and making it very clear what the contribution of the research is. It was unclear in the paper what the overall contribution is to the profession.

Response 18: The purpose and contribution of this study have been rewritten at the end of Introduction in line 49.

Point 19: Line 100 – Provide a better transition between sections.

Response 19: The transition between sections has been rewritten in line 101. And the data sources section has also been rewritten.

Point 20: Line 102 – Use “three” in place of “3.”

Response 20: The mistake has been revised in line 104.

Point 21: Line 110 – No “the” needed before “Caltrans.”

Response 21: The data sources section has also been rewritten. This mistake has no longer existed.

Point 22: Line 112 – Rewrite as “The injury collisions in this study are injury…and the non-injury collision is a property damage only (PDO) collision.” Define PDO here as an acronym and then use just the acronym later on.

Response 22: The sentence has been rewritten and PDO has been defined in line 113.

Point 23: Line 114 – Use LOS, not Los. This is applicable throughout the remainder of the paper as Los is used quite often (more than LOS).

Response 23: The acronym Los has been replaced by LOS in all cases.

Point 24: Line 115 – Once an acronym is defined (in this case LOS), use the acronym consistently throughout the remainder of the document. This is the case in a couple of locations in this paragraph.

Response 24: The acronym has been revised in all cases, such as LOS, MCMC, etc.

Point 25: Line 117 – Refer to the tables by number. It is also best to refer to the table before the table appears in the document. In that case, this paragraph would come before the table, rather than after the table.

Response 25: The number of the table has been added in line 115. And the table 1 has been moved to the back of this paragraph.

Point 26: Line 118 – Again, refer to the table by number. This paragraph overall (and several others in the document) is quite choppy. It jumps around quite a bit. Look at ways to make the writing overall flow better.

Response 26: The number of the table has been added in line 117. And this paragraph has been rewritten.

Point 27: Line 127 – The word “mode” here seems like it should be “model.” This is the case throughout this paragraph.

Response 27: The mistake has been revised in line 174 and all the other cases.

Point 28: Line 134 – How does Figure 1 show the absolute probability of collision by different types and severities can be calculated by all these equations? It is unclear to me how this “shows” the equations. You need a better connection between the figure and the equations along with a better discussion on both to help the reader understand the process better.

Response 28: Figure 1 shows the framework of five-stage Bayesian random parameter sequential logit model, the related sentence has been added in line 165. Specifically, the different stages of model have been introduced in line 173. Finally, the absolute probability of collision by different types and severities have been calculated by equation (8)-(13).

Point 29: Line 141 – Remove the word “metric” as it seems that “geometric” is sufficient. What other variables could be included? Rather than use “etc.” include an exhaustive list if possible.

Response 29: The transition between sections has been rewritten in line 139. This mistake has no longer existed.

Point 30: Line 169 – MCMC is already defined as an acronym.

Response 30: The mistake has been revised in all case.

Point 31: Line 177 – Change “assume” to “assumed.”

Response 31: The sentence has been rewritten in line 152. Therefore, the mistake has no longer existed.

Point 32: Line 181 – It seems that “mode” here should be “model”?

Response 32: The section 4.2 has been rewritten. Therefore, the mistake has no longer existed.

Point 33: Line 183 – Be more specific on where the collision types and severities were calculated. Include equation numbers and clarify. Overall the discussion of the model is quite confusing.

Response 33: The section 4.2 has been rewritten.

Point 34: Line 187 – References 23 through 31 are all lumped together as part of one sentence. That is a lot of references to tie into one sentence. More detail on this should be provided.

Response 34: The references have been introduced in detail in line 157.

Point 35: Line 189 – Should this refer to Figure 1? It does not make sense to reference Figure 2 at this point in the document.

Response 35: The mistake has been revised in line 166.

Point 36: Line 191/192 – In line 191 you refer to “three random parameters” and then in line 192 you say “These seven random parameters…” Is it three or seven? Or do these even go together? This needs to be clarified.

Response 36: This problem has been clarified in line 165. In detail, the five-stage Bayesian random parameters sequential logit model has five stages. Four Bayesian random parameters binary logit models were developed in the stage 1-4. Three Bayesian random parameters binary logit models were developed in the stage 5. Therefore, the total number of the Bayesian random parameters binary logit models is seven. These seven Bayesian random parameters binary logit models make up the five-stage Bayesian random parameters sequential logit model.

Point 37: Line 217 – Instead of “with over lines…” you could refer to them as bars.

Response 37: The section 4.2 has been rewritten. Therefore, the mistake has no longer existed.

Point 38: Line 221 – Provide an introductory paragraph to the subsections. Also, the word “od” should be “of” I believe. Be sure to proofread and spell-check.

Response 38: The introductory paragraph has been added in line 185. And the title of section 5 has been revised in 184. Therefore, the mistake has no longer existed.

Point 39: Line 223 – Always use a capital letter for the LOS designations (in this case “E.”

Response 39: “E” has been removed in line 191.

Point 40: Line 225 – Use the acronym LOS in place of “level of service” once it has been defined.

Response 40: The mistake has been revised in all case.

Point 41: Line 227 – Add “the” after “…highly correlated with…”

Response 41: “the” has been added in line 196.

Point 42: Line 229 – Use LOS throughout the table. What is the baseline value for LOS A? It is unclear and the discussion compares everything with LOS A but it is never clarified what that value is.

Response 42: LOS has been used throughout the table in all case. In this section, as shown in Table 4, every LOS is a binary variable. LOS A is regarded as the reference level of Bayesian conditional logit model. Thus, there is no value for LOS A in the results of models. The Bayesian conditional logit model was used to explore the relative safety performance between LOS A and other LOS. The explanatory sentence has been rewritten in line 191. In previous studies, there is also no value for the reference level in Bayesian conditional logit model. The reference is as follow:

Xu, C.; Liu, P.; Wang, W.; Li, Z. Safety performance of traffic phases and phase transitions in three phase traffic theory. Accident Analysis and Prevention. 2015, 85, 45-57.

Point 43: Line 234 – Remove “the” at the end of this line.

Response 43: The sentence has been rewritten in line 200. Therefore, the mistake has no longer existed.

Point 44: Line 242 – The discussion provided for the “hit object collision” does not make sense. I see similar odds ratios for all LOS. How is LOS B and LOS C “significantly greater than LOS A”? LOS E has a higher odds ratio than LOS B so this doesn’t make sense. There is some confusion in this discussion.

Response 44: In the result of table 3, the 2.50% value and 97.50% value are both positive for LOS B and LOS C. Thus, it implies that the odds ratio of LOS B and LOS C are significantly greater than LOS A. In contrast, the 2.50% value and 97.50% value of LOS D, LOS E, and LOS F are negative and positive, respectively. Thus, it indicates that he odds ratio of LOS D, LOS E, and LOS F are not significantly greater than LOS A. The analysis procedure can be found in previous studies. The reference is as follow:

Xu, C.; Liu, P.; Wang, W.; Li, Z. Safety performance of traffic phases and phase transitions in three phase traffic theory. Accident Analysis and Prevention. 2015, 85, 45-57.

Point 45: Line 270 – Rewrite as “In addition to LOS, five other candidate variables including…were also taken into consideration at every stage.”

Response 45: The sentence has been rewritten in line 237.

Point 46: Line 274 – Provide a better transition between section 4.2 and the subsections that follow it.

Response 46: The transition paragraph has been rewritten at the beginning of section 4.2 in line 229.

Point 47: Lines 276-280 – Which columns are you comparing here? The results do not match what is in the table. Later on, you always refer to the mean values as the comparison. The coefficients on the mean values are always negative so how can some probabilities increase and some decrease? The discussion does not match the results. This is also true in the stage 2 model discussion. Please review all discussions and update accordingly.

Response 47: In table 4, the candidate variables in models have been explained to better analyze the model results.

Point 48: Line 298 – This section has a better discussion that matches what the tables show. One thought in this section is to only use the acronym PDO as it should have been defined several sections back.

Response 48: The section 4.2.2 has been discussed and rewritten in detail. And the acronym PDO has been used instead of property damage only in all case.

Point 49: Line 314 – The conclusions and discussion section should first clearly restate the purpose and need, identify how the purpose and need were addressed, and then provide a discussion of the results. There are several changes to make with the English in this section including changing “establish” to “established” in Line 316, “analyzing” to “analyze” in Line 317, “predict” to “predicted” in Line 320, and “predict” to “predicted” in Line 321. The conclusions need to also make it very clear what the contribution is. I do not walk away from this paper with a true understanding of the overall contribution that this research makes to the literature. This is a critical change that needs to be made.

Response 49: The purpose and need is introduced in line 293. The methods how the purpose and need were addressed are introduced in line 296. The paragraph 2-3 of conclusion section are used to discuss the results in line 301 and line 312. The paragraph 4 of conclusion section is used to introduce the overall contribution of this research in line 319. The paragraph 5 of conclusion section is used to introduce the limitation of this research in line 323.

Reviewer 2 Report

The manuscript aims to identify how the level of service (LOS) impacts the crash types and severities. My specific comments are listed as follows.

  1. Page 2, Line 82. The article of the reference 15 was not written by one of the authors. Please illustrate it.
  2. In Section of Introduction. The authors would better illustrate the specific research gap of the study.
  3. In Section of Data. How did the authors select the case and control sample?
  4. In Table 4 and Table 5. What is the baseline for level of service (LOS) in the model of different stages? Could you discuss the model estimation result?
  5. In Section 5. The model discussion part would better be illustrate in details in a separate section before the Section of Conclusion.
  6. In Figure 2. Please illustrate the reason why the LOS F has the highest injury collision probability.
  7. Lines 335-341. In the model discussion, please illustrate the baseline of the contributing factors for crash impact analysis.
  8. Lines 346-347. What do the results mean in the practice of traffic safety management?
  9. Grammar errors need to be thoroughly checked through the manuscript. For example, Line 316-317. “establish to analyzing” should be revised as “established to analyze”.

Author Response

Response to Reviewer 2 Comments

Point 1: Page 2, Line 82. The article of the reference 15 was not written by one of the authors. Please illustrate it.

Response 1: The mistake has been revised in line 87.

Point 2: In Section of Introduction. The authors would better illustrate the specific research gap of the study.

Response 2: The specific research gap of the study has been rewritten in the sentence in line 43.

Point 3: In Section of Data. How did the authors select the case and control sample?

Response 3: The interpretation of the case and control sample is introduced in line 109.

Point 4: In Table 4 and Table 5. What is the baseline for level of service (LOS) in the model of different stages? Could you discuss the model estimation result?

Response 4: The baseline for level of service (LOS) and other variables have been shown in Table 4. The model estimation results have been discussed in section 5.

Point 5: In Section 5. The model discussion part would better be illustrate in details in a separate section before the Section of Conclusion.

Response 5: The model discussion has been rewritten in details in the section 5 (Results and Discussion) before the section of conclusion.

Point 6: In Figure 2. Please illustrate the reason why the LOS F has the highest injury collision probability.

Response 6: The reason why the LOS F has the highest injury collision probability has been explained in line. In addition, the results in Figure 2 is a conclusion of the results in Table 3.

Point 7: Lines 335-341. In the model discussion, please illustrate the baseline of the contributing factors for crash impact analysis.

Response 7: The baseline for level of service (LOS) and other variables have been shown in Table 4.

Point 8: Lines 346-347. What do the results mean in the practice of traffic safety management?

Response 8: The overall contribution of this research has been rewritten in line 319. The sentence in original version has been removed.

Point 9: Grammar errors need to be thoroughly checked through the manuscript. For example, Line 316-317. “establish to analyzing” should be revised as “established to analyze”.

Response 9: The grammar errors have been revised in all case.

For example:

In line 297, “establish to analyzing” has been revised as “established to analyze”.

In line 18, the word “object” has been revised as “objective.

In line 20, “…been establish…”has been revised as “…been established…”

In line 21, the word “analyzing” has been revised as “analyze.”

In line 28/76/82, the word “significantly” has been revised as “significant”

In line 64, the word “sates” has been revised as “states”.

In line 149//153, “logistic” has been revised as “logit”.

Reviewer 3 Report

This is an interesting paper that further explore relations between traffic flow states and LOS on one side, and safety on the other.

The paper needs some editing for English. Make sure that certain words are used consistently (e.g. logistic models -> logit models)

The first section needs to be separated in two, Introduction and Literature review. In the introduction, you need to briefly introduce the paper, data and methodology, as well as the goal of the paper and the significance/contributions of your research. All the reviews you performed should be moved to the Lit review section.

The same goes for the Conclusions and Discussion section, it would flow much better to separate those into two.

Author Response

Response to Reviewer 1 Comments

Point 1: The paper needs some editing for English. Make sure that certain words are used consistently (e.g. logistic models -> logit models)

Response 1: The grammar errors have been revised in all case.

For example,

In line 149//153, “logistic” has been revised as “logit”.

In line 297, “establish to analyzing” has been revised as “established to analyze”.

In line 18, the word “object” has been revised as “objective.

In line 20, “…been establish…”has been revised as “…been established…”

In line 21, the word “analyzing” has been revised as “analyze.”

In line 28/76/82, the word “significantly” has been revised as “significant”

In line 64, the word “sates” has been revised as “states”.

Point 2: The first section needs to be separated in two, Introduction and Literature review.

Response 3: The first section has been separated into two sections (Introduction and Literature review).

Point 3: In the introduction, you need to briefly introduce the paper, data and methodology, as well as the goal of the paper and the significance/contributions of your research.

Response 3: In the section 1 (Introduction), paragraph 3 in line 49 is used to introduce the paper, including the purpose, data, methodology and the contributions of this study.

Point 4: All the reviews you performed should be moved to the Lit review section.

Response 4: All the reviews have been moved to the section 2 (literature review).

Point 5: The same goes for the Conclusions and Discussion section, it would flow much better to separate those into two.

Response 5: The discussion section has been removed to section 5 (Results and Discussion). The conclusion section has been removed to section 6 (Conclusion).  Both sections have been rewritten.

Round 2

Reviewer 1 Report

Modeling Collision Probability on Freeway: Accounting for Different Types and Severities in Various LOS

The purpose of this paper is to identify how level of service affects collision types and severities. As with the original submittal, the paper has some interesting relationships outlined that would provide good information for practitioners and researchers. Although several changes have been made to the paper, there are still several changes that should be made to the paper before it is considered for publication, especially in those sections that were added to the paper.

The following comments are provided listed by line number:

Line 17 – Use “I-880” consistently throughout the paper (not I880). Also, note where in California this is located (California is a very large state!).

Line 28 – The word “the” should be removed before “significant.” As noted in my original review, this line should read “…number of lanes have significant effects on…”

Line 42 – Here, and in several places in the document, you use “etc.” to note more factors. Do you know all the factors? It is best to include all, rather than using etc. The phrase etc. should be used very sparingly.

Line 43 – Change “divided” to “dividing” at the end of this line.

Line 45 – The term “Especially” to start off this sentence is confusing. Not sure what it means here.

Line 47 – Change “significantly” to “significant” and “different” to “difference” so that this line would read “…there is a significant difference in the safety performance…”

Line 52 – Change “I880” to “I-880” here and throughout the document. Also note where this is located (in this line you just say the United States – previously you said California, but you should be even more specific and note the city in California).

Line 61 – You start out this section noting that “most” studies have explored the collision mechanism… but then only have two references. It seems that “most” would be more than two. The word “have” should be added in to what you have in the text to get “…most studies have explored…”

Line 63 – Change “It indicated…” to “The researchers indicated…”

Line 69 – Rewrite to say “…is insufficient qualitative analysis…”

Line 71 – Add “a” after “…flow states on…”

Line 74 – Change “condition” to “conditions”

Line 77 – remove “the” before “sideswipe collisions.”

Line 80 – Change “has” to “is” to read “…collision is significantly related to occupancy…”

Line 87 – Change this sentence to read “…collisions are significantly different for varying traffic flow states.”

Line 92 – Be sure to spell check. Numberous is not a word. It should be “Numerous.” This sentence is also a little bit confusing. How can traffic states be classified into different traffic states? It seems that they are already in different traffic states. Reword this sentence to make it more understandable.

Line 96 – I think that reference 14 here should be 13 and reference 15 should be 14. Be sure to check all references. Reference 15 doesn’t seem to ever actually be used. It doesn’t match anything you have here.

Line 102 – Overall this paragraph is very choppy. It needs to be reworded to flow more smoothly. Some specific changes to this sentence would be to say, “The segment then of the I-880 freeway in California, United States is 34 miles.”

Line 103 – Avoid beginning a sentence with a number. If you do, it must be spelled out.

Line 106 – For units of measurement, always use the number. In this case it would be “5 minutes to 10 minutes.” The end of this sentence should be rewritten as well to add in “…occurrence time of the reported collision.” There are a lot of grammatical changes that need to be made throughout the entire document.

Line 108 – The word “proved” should be “proven.” In the next sentence, you need to start with “The non-collision case…”

Line 109 – If you are going to use military time (e.g., 15:00) you do not need to include “pm.” For dates, do not include “th” after the date, simply say Nov. 20, 2009.

Line 112 – How is the ratio between collision and non-collision calculated? This is very confusing to the reader and needs a better explanation. Also, what does this ratio mean? What is its significance? Why is it important to include?

Line 113 – Simply define PDO when you introduce the crashes. Rewrite the sentence as “…injury or fatal collision, and property damage only (PDO) collision.

Line 138 – What does this mean? “Therefore, the calculate equation of probability…” does not make sense.

Line 139 – Rewrite slightly “…for this model using Markov Chain Monte Carlo (MCMC) methods.”

Line 140 – Add “the” before “…model have a prior distribution.”

Line 150 – Remove “the” before “…analyzing collision severities…”

Line 152 – Change “parameters estimates…” to “parameter estimates…”

Line 156 – Remove “can” before “…explains the correlation…”

Line 162 – Remove the “s” in “results”

Line 185 – The two sentences here need to be combined; they are repetitive.

Line 198 – Reword slightly to say “…a total of 10000 iterations, 4000 burn-in iterations, and three parallel MCMC chains for Bayesian inference.”

Line 199 – Combine the last sentence here and the first full sentence on Line 200 to read “The results of the Bayesian conditional logit models are shown in Table 3.” Always look for ways to simplify the writing.

Table 3 – “MC errror” should be “MC error” – always be sure to spell check.

Line 204 – There are two commas after “Specifically…”

Line 206 – Although changes have been made here it is still not explained sufficiently. What is it that tells the reader the level of significance? Make this very clear. You note which LOS are significantly greater than LOS A but don’t explain the way you come to this conclusion. In looking simply at Odds ratios, all the Odds ratios are similar in magnitude. You need to explain the negative percentiles and how this tells you the significance. It is still not explained well.

Line 216 – Again, spell check! “manervers” is not a word.

Line 219 – Again, more spell check and proofreading, “somne” is not a word. Change “significantly” to “significant” here as well.

Line 231 – Remove “the” before “section 2.4.1” as well as before “stage” when discussed. This is something that needs to be changed throughout this section. Too many “the” included here.

Line 239 – Change “…the section 4.1” to “…that of section 4.1.”

Line 241 – Change the title to remove “The” from the section title “Sequential model for collision types”

Line 242 – Change “stages 1 to stage 4” to “stage 1 to stage 4.”

Line 244 – Change “In previous studied…” to “In previous studies…”

Line 247 – What is meant by “still related high than congestion…”?

Line 254 – Be consistent throughout in your use of commas in lists. For the most part you have a comma for each item in your list but here you start to not include the comma before the last item in the list.

Line 257 – Change “conditional” to “condition”

Line 263 – There is an extra “L” left over from the old Los.

Line 266 – Change to read “…was established to predict rear end collision likelihood.”

Line 268/269 – Remove “the” before “…rear end collision probability” and “positive effects on…”

Line 269 – Change “In LOS…” to “For LOS…” and remove “situation” from the text.

Line 271 – Remove “The” in the table title.

Line 272 – Remove “The” in the section title.

Line 275 – Change “indicates” to “indicate.”

Line 276/277/278 – Change “collision” to “collisions.”

Line 279 – Change “indicated” to “indicate” and “conditional” to “condition.” The phrase “less space in few number of lanes” is also confusing – not sure what is meant here.

Line 284 – Change to read “…fewer numbers of lanes…”

Line 286/290 – Spell check again, “occours” is not a word.

Line 291 – Remove “The” from the table title.

Line 294 – Change “affects” to “affect”

Line 295 – Clearly identify where I-880 is and be sure to use the dash in I-880.

Line 313 – Change to read “…have significant effects on different collision…” “It was found that fewer lanes and low visibility can both increase…”

Line 315 – Should “Ramp” be “Ramp spacing”?

Line 316 – Change “conditional” to “conditions”

Line 323 – Add “the” after”…to be studied in…” Change “Firstly” to “First…”

Line 326 – Spell check, “vertified” is not a word.

Again, there is a lot of interesting information in this paper, but the presentation of the paper still needs work in order to be publishable. Please continue to work on this and have the paper proofread by an English Editor to help get it to the point where it is ready to be published.

Round 3

Reviewer 1 Report

Thank you for your revisions. There are still minor grammatical changes that should be applied but can likely be handled by the Editor should the paper be accepted for publication. A few specific changes would include:

Line 213 - The reference to Table 3 here is repetitive as it was just noted in the previous line.

Line 274 - I still believe "weather conditional" should be "weather conditions" This was changed in many places but not here.

Line 328 - When referring to I-880 it is located "in" California and the "United States" rather than "United State."

Thank you for your research, it is very interesting.

Author Response

Dear reviewer:

We would like to thank the reviewer for thoughtful review of our manuscript. We believe that the additional changes we have made in response to the reviewer comments have made this a significantly stronger manuscript. Below is our point-by-point response to the reviewer’s comments:

Point 1: Line 213 - The reference to Table 3 here is repetitive as it was just noted in the previous line.

Response 1: The repetitive sentence has been removed from line 213.

Point 2: Line 274 - I still believe "weather conditional" should be "weather conditions" This was changed in many places but not here.

Response 2: "weather conditional" has been changed to "weather conditions" in Table 4.

Point 3: Line 328 - When referring to I-880 it is located "in" California and the "United States" rather than "United State."

Response 3: “in” has been added before “California” in line 328 and "United State" has been changed to "United States" in line 329.
